# Effectiveness of Simulation-Based Empathy Enhancement Program for Caregivers (SEE-C) Evaluated by Older Adults Receiving Care

**DOI:** 10.3390/ijerph18157802

**Published:** 2021-07-23

**Authors:** Kyuwon Lee, Areum Han, Tae Hui Kim

**Affiliations:** 1Industry Academic Cooperation Foundation Office, Yonsei University, Mirae Campus, 79, Ilsanchogyo-gil, Wonju 26425, Korea; otssh101@gmail.com; 2Department of Occupational Therapy, University of Alabama at Birmingham, SHPB 340, 1720 2nd Ave South, Birmingham, AL 35294, USA; ahan@uab.edu; 3Department of Psychiatry, Yonsei University Wonju College of Medicine, Wonju 26426, Korea

**Keywords:** empathy, simulation training, elderly, aged, caregivers, quality of health care, patient reported outcome measures

## Abstract

The aim of this study was to examine whether a Simulation-based Empathy Enhancement program for Caregivers of the Elderly (SEE-C) was effective in increasing program satisfaction and positive emotional changes of older adults. A total of 100 older adults living alone were randomly assigned to experimental and control groups. The experimental group was interviewed by caregivers who experienced SEE-C while the control group was interviewed by caregivers who did not experience SEE-C. In both elderly groups, post session satisfaction and affective state were assessed using a Session Evaluation Questionnaire (SEQ). Chi-square test and Mann-Whitney U test were conducted. The experimental group (*n* = 49) reported significantly higher scores than the control group (*n* = 51) for all three categories of SEQ: session-depth (Mann-Whitney U = 1651.5, *p* = 0.005), session-smoothness (Mann-Whitney U = 1803.0, *p* = 0.000), and emotion-positivity (Mann-Whitney U = 1783.0, *p* = 0.000). However, the experimental group had significantly lower scores for the arousal category of SEQ (Mann-Whitney U = 873.5, *p* = 0.009). SEE-C could have a positive impact on interviews for elderly care in terms of raising the satisfaction of the interviewee.

## 1. Introduction

Living alone is potentially a major risk factor for depression and suicide in older adults [1,2]. The elderly who are living alone not only have lower physical functions and life satisfaction levels than those who live with family, but also have higher levels of perceived stress and lower levels of perceived health, making them more vulnerable to psychological and social aspects [3]. For this reason, the Korean government has hired direct care workers called “life managers of elderly people living alone” to check the safety of the elderly living alone and to reduce social isolation, providing regular check-in services through home visits once a week and telephone [4]. Care recipients are selected through physical, mental, and economic vulnerability assessments. Approximately 30% of the elderly living alone are receiving such services. Caregivers are hired without special qualifications. Each caregiver cares for about 20 seniors per year. Although these employed caregivers are aware of their roles for emotional support of the elderly, they have a high demand for education and resources because they find difficulties in building rapport with the elderly living alone and in providing emotional support [5].

One way to support caregivers with above-mentioned difficulties is by empowering empathy of caregivers [6,7]. Empathy is “*the unique capacity of the human being to feel the experience, needs, aspirations, frustrations, sorrows, joys, anxieties, hurt, or hunger of others as if they were his or her own*” [8]. High levels of empathy are associated with healthy relationships and prosocial behaviors [9]. Studies have emphasized the need to provide education for care service providers to improve their empathy [10,11,12,13]. According to a meta-analysis of randomized controlled trials on the efficacy of empathy training, empathy training tends to be effective in improve the empathy of caregivers, although more experimental research is warranted to understand the impact of empathy training on different types of trainees and outcomes [14].

Recently, simulation-based training has emerged as an important means to educate behavioral skills and knowledge relevant to empathy [15]. Simulation-based experience and debriefing training provide an opportunity to develop and explore empathy-based communication skills [16]. The authors of this study have developed a Simulation-based Empathy Enhancement Program for Caregiver of the Elderly (SEE-C) and compared a simulation-based empathy training group to a lecture-based empathy training group enrolling 209 social workers and direct care workers of older adults living alone. The authors have found that SEE-C can improve the empathy and reduce the compassion fatigue of social workers who perform administrative tasks and provide care service for general older adults including elderly living alone in community welfare centers [17]. On the other hand, among direct care workers who visited the home of the elderly living alone, the lecture-based empathy training group had significantly higher levels of empathy than the simulation-based group, although pretest-posttest differences were found only for the lecture-based group [18].

Results of our previous studies may indicate that teaching methods of empathy education should be different depending on roles and primary tasks of care providers. In addition, outcome measure selection by using self-reported empathy measures has limitations. Empathy is a multidimensional construct involving cognitive, emotional, moral, and behavioral aspects [19]. Self-reporting bias might occur when using self-reported empathy measures. In addition, behavioral empathy cannot be measured by self-reporting [20,21]. For example, studies have found that physicians’ self-assessed empathy is not correlated with patients’ perception. All of these findings suggest the need to measure efficacy of an empathy training program for care providers of older adults from perspectives of older adults [22].

Very few studies have been conducted on non-medical caregivers who care for the elderly. In case of non-medical caregivers in Korea, emotional labor during work has a direct effect on burn out with a negative effect on job satisfaction, which is related to the lack of empathy-related education for them [23]. Therefore, if empathy-related education is provided to them, it can be expected to have a positive impact on caregiver’s burnout and job satisfaction, which can also have a good impact on the satisfaction of care recipients.

The aim of this study was to measure the effectiveness of a simulation-based empathy enhancement program for direct care workers of older adults living alone by evaluating older adults who received home visiting care by trained direct care workers through the simulation-based empathy enhancement program.

## 2. Materials and Methods

### 2.1. Research Design

All older adults, who participated in the present randomized controlled trial, were randomly assigned into experimental and control groups. The experimental group was interviewed by direct care workers who were trained through the SEE-C while the control group was interviewed by workers who were not trained through the SEE-C. Direct care workers were women working in local senior welfare centers. There were no significant differences in age, education, or length of work between groups who received SEE-C education and those who did not. Older adults did not know whether their interviewers received SEE-C or not. The interview procedure was divided into four sessions based on autobiographical memory. Interviews were conducted in homes of the elderly for an hour.

This study was part of a larger research project for developing a semi-structured interview protocol to talk about autobiographical memory for the purpose of supporting emotions of the elderly living alone. The authors have developed a Simulation-based Empathy Enhancement program for Caregivers of the Elderly (SEE-C) by modifying the Dementia LiveTM program of the AGEu-cate Training Institute in the USA [24] and the Korean Dementia Simulation program for Caregivers [25,26], trying to incorporate it into an interview protocol to talk with the elderly living alone. SEE-C was modified by changing focus from dementia to aging and by adding a brief breathing meditation session [17,18]. It aimed to help care providers enter the world of older adults by experiencing similar feelings and challenges from perspectives of older adults. It also aimed to help care providers cultivate mindfulness as a potential method for reducing stress while increasing empathy [27].

### 2.2. Sample Size Calculation and Recruitment

The G*power software ver. 3.1.9.4 (Heinrich-Heine-Universität Düsseldorf, Düsseldorf, Germany) was used to calculate the required sample size of older adults. With a significance level of 5%, a medium effect size of 0.5 (Cohen’s d), and a power of 70%, the required sample size per group was 53. Among those who participated in the 2018 national survey of elderly people living alone in Wonju, older adults were selected as samples for the present study after excluding those who were receiving public care service because they were relatively healthy without having the lowest income level. Five administrative districts were selected for convenience. A total of 372 adults aged between 65 and 80 years were contacted by telephone and mail. We recruited 144 elderly people living alone and randomly assigned them into experimental and control groups. Data collection and interview implementation were done between 4 May 2019 and 25 June 2019. The analysis was conducted for 100 older peoples who answered the assessment without any errors or omissions (Figure 1).

### 2.3. Ethics

This study was conducted after obtaining approval from the Institutional Review Board (IRB) of Yonsei University Wonju Severance Chirstian Hospital, South Korea (approval number: CR 318026). Researchers explained the research purpose, benefits, and risk to all participants and obtained informed consent.

### 2.4. Measurement

#### 2.4.1. Health Status Evaluation

In this study, the health status of each subject was measured using the Korean version of Euro Quality of Life Questionnaire 5-Deminsional Classification three-level version, EQ-5D-3L [28]. EQ-5D-3L is a health-related quality of life measurement tool. It includes five items to measure five dimensions of health: mobility, self-care, usual activities, pain/discomfort, and anxiety/depression. The closer the total score is to 1, the better the health [29]. We also used Korean Version of the Revised UCLA Loneliness Scale; RULS [30] to find out the degree of loneliness of the elderly. Mini-Mental State Examination for Dementia Screening (MMSE-DS) [31] is one Korean version of screening tools for cognitive impairment. Older adults with MMSE-DS score below −1.5 standard deviation were excluded. The Korean version of Short form of Geriatric Depression Scale (SGDS-K) [32] was used to exclude elderly people suspected of depression.

#### 2.4.2. Interview Session Evaluation

Session Evaluation Questionnaire (SEQ) was developed to measure the impact of an interview [33]. It was used in this study to evaluate the satisfaction and emotions of older adults after a visit interview. The Korean version of SEQ consisting of 19 items was used [34]. A total of 19 pairs of bipolar adjectives are presented with a 7-point scale. The SEQ was divided into session evaluation section and post session mood section. The session evaluation section consists of categories of ‘depth’ and ‘smoothness’. The post-session mood section consists of categories of ‘emotion-positivity’ and ‘emotion-arousal’. The interviewee gave high marks to question items in the ‘depth’ category of the SEQ if they thought that the time spent with the interviewer was valuable. They gave high marks to items in the ‘smoothness’ category if they felt comfortable and stable during the session. A high score of emotion-positivity after session indicated that the interviewer gave confidence and clarity to the interviewee and made the interviewee feel happy. The arousal score refers to feeling active and excited. However, there are things to consider about arousal. In a SEQ validation study for utilization in Korea, several questions indicating awakening were opposite to the United States. Such differences may be because of cultural or linguistic differences. Therefore, a follow-up study will be needed on awakening [35].

### 2.5. Statistical Analysis

Collected data were analyzed using SPSS version 25 (IBM Corp., Armonk, NY, USA). Descriptive statistics were used to summarize general characteristics of participants. The Kolmogorov-Smirnov test was used to test the normality of data. Because data were not normally distributed, Chi-squared test (χ^2^ test) for categorical variables and Mann-Whitney U test for continuous variables were used to determine the significance of differences in general characteristics of participants.

## 3. Results

Demographic characteristics of 100 elderly people living alone (49 in the experimental group and 51 in the control group) are shown in Table 1. The average age of these subjects was 72.1 ± 4.03 years. Of these subjects, 63% were women. The average educational background was 7.2 ± 3.86 years and the average period of living alone was 19.4 ± 12.56 years. There was a statistically significant difference in age (mean) between the two groups, but the median of the control group was 70 and the median of the experimental group was 74. Also, the quartile range was 6, which was same for both groups.

Table 2 and Figure 2 show differences in average scores of the SEQ between the two groups. Among four categories of SEQ, the experimental group reported significantly higher scores than the control group for three categories: session-depth (Mann-Whitney U = 1651.5, *p* = 0.005), session-smoothness (Mann-Whitney U = 1803.0, *p* = 0.000), and emotion-positivity (Mann-Whitney U = 1783.0, *p* = 0.000). On the other hand, the experimental group had significantly lower scores for the emotion-arousal subscale (Mann-Whitney U = 873.5, *p* = 0.009) than the control group.

## 4. Discussion

This study was designed to investigate the effectiveness of a simulation-based empathy enhancing program for caregivers of the elderly (SEE-C) by directing evaluating satisfaction and emotion of older adults using SEQ. Older adults were randomly assigned to either the experimental group or the control group. The experimental group showed significantly higher scores for three SEQ categories (depth, smoothness, and emotion-positivity) but significantly lower scores for the emotion-arousal category of SEQ than the control group.

Depth and smoothness as categories of SEQ evaluate the interview time. Thus, a deep and smooth feeling means a high satisfaction with the interview [35]. If post-session emotion-positivity is significantly higher, it means that there is a positive change in mood due to the interview [35]. However, in this study, only arousal showed a different direction from the other three categories. The arousal of the control group was significantly higher than that of the experimental group. Such result is expected because a previous study has reported that arousal is increased when interview is not smooth or felt negative [34].

This difference in satisfaction between the two groups for the same structural interview in this study can be seen as a result of increased empathy for the elderly in real-world situations due to the provision of simulation-based training programs [36]. In one study, when 150 medical students were given simulated auditory hallucination for 40 min to determine how it influenced the improvement of understanding and empathy for mentally ill patients, significant improvement in empathy of medical students was observed [37]. One study has also shown that the understanding and empathy for the elderly are improved when nursing and nutrition students (*n* = 127) are provided with Aging games to experience physical disabilities caused by aging [38].

In most of previous studies, service providers measured their empathy levels by themselves. What was noteworthy of the present study was that older adult who were service recipients measured their experience to reflect empathy levels of service providers. Such self-assessment commonly used in many studies may not be well correlated with the reality generally observed by others. Education and expectations about what is considered a desirable attitude for caregivers may affect the cognitive way they appreciate themselves. However, it does not guarantee a change in behaviors that takes empathetic attitudes toward others [17].

Several studies on empathy training were mainly conducted on medical staff or medical students. The empathic attitude of care providers is an essential factor for improving the quality of care. The demand for empathy is steadily increasing not only in the medical field, but also in various health workers [11,12,22]. Empathy-based communication skills between caregivers and recipients can lower stress levels of both sides and further reduce the incidence of problems associated with psychological and health [9]. Therefore, empathy training and empathetic care can be used to create a symbiotic relationship, not just a one-sided sacrifice of caregivers for care reciprocal, by smooth interaction and emotional or behavioral change. Various studies on empathy are needed for geriatric care professionals and non-medical caregivers who care for the elderly in the future. With increasing numbers of old people, dementia patients, and elderly living alone, the number of people taking care of patients is bound to increase. Thus, policy interest in community care for the elderly is very high.

This study has some limitations, making it difficult to generalize results of this study because it only targets the elderly living alone in a limited area. In addition, the number of samples was slightly smaller than the calculated sample size. Moreover, effects of age of the two groups were not corrected through a nonparametric statistical analysis. However, this study identified the short-term effect of SEE-CE through a random design. In the future, we need to use a larger sample size to determine the long-term effect of SEE-C. This study did not include an understanding or approach to evaluate empathy based on neuroscientific evidence. However, empathy is a multi-dimensional complex concept [12,39]. It is not easy to define or measure. Research on scientific approaches for measuring emphathy is quite lacking [19,40]. Therefore, in the future, various empathy promotion programs and evaluation methods for various objects should be developed and studied. Continuous research on neuroscientific changes related to empathy is also needed.

## 5. Conclusions

In conclusion, this study found that the elderly cared by caregivers who participated in SEE-C had higher session satisfaction and positive emotion change based on evaluation by the elderly themselves. This suggests that SEE-C is effective in promoting empathy of caregivers for the elderly and providing a better level of response in the real field.

## Figures and Tables

**Figure 1 ijerph-18-07802-f001:**
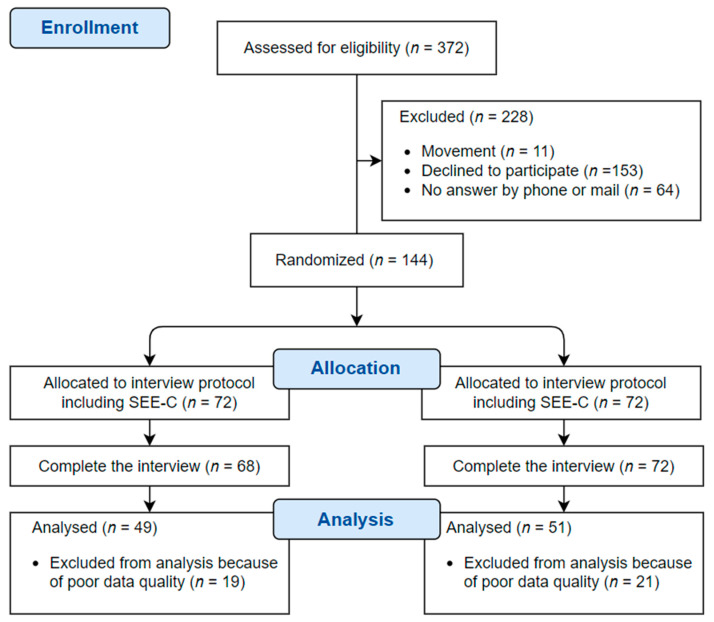
CONSORT Flow Diagram.

**Figure 2 ijerph-18-07802-f002:**
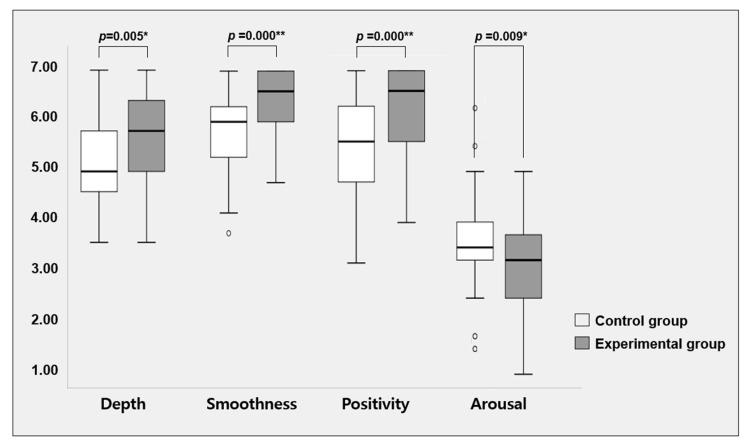
Scores for four subcategories of SEQ. Data are shown as a box and whisker plot. In this plot, whiskers extend to the largest and smallest data points and the box extends from the upper quartile to the lower quartile crossed by a line at the median of data. Subjects in the Experimental group were interviewed by direct care workers who experienced SEE-C. Subjects in the Control group were interviewed by direct care workers who did not experience SEE-C. * Statistically significant at *p* < 0.01. ** Statistically significant at *p* < 0.001.

**Table 1 ijerph-18-07802-t001:** General characteristics of participants in the experimental group and the control group.

Category	Experimental Group (*n* = 49)	Control Group (*n* = 51)	*p*-Value
Age(years)	73.02 ± 4.0	71.16 ± 3.8	0.020 *
Gender			
Male	19 (38)	18 (36)	0.718
Female	31 (62)	32 (64)	
Years of Education	6.78 ± 3.9	7.24 ± 4.1	0.729
Years of Living Alone	17.94 ± 11.7	20.75 ± 13.2	0.313
EQ-5D-3L	0.82 ± 0.2	0.83 ± 0.2	0.807
RULS	37.65 ± 10.2	39.75 ± 9.6	0.386
MMSE-DS	27.94 ± 2.4	27.90 ± 1.9	0.538
SGDS	2.45 ± 2.5	3.08 ± 2.9	0.208

Note. Data are presented as mean ± standard deviation, number (percentage), or *p*-value for Mann-Whitney U test and Chi-square test. EQ-5D-3L: Euro Quality of Life Questionnaire 5-Deminsional Classification three-level version; MMSE-DS: Mini-Mental Status Examination for Dementia Screening; RULS: Revised UCLA Loneliness Scale; SGDS: Short form of Geriatric Depression Scale. * Statistically significant at *p* < 0.05.

**Table 2 ijerph-18-07802-t002:** Average SEQ scores for the experimental group and the control group.

Categories of SEQ		Experimental Group (*n* = 49)	Control Group (*n* = 51)	*U*-Value	*p*-Value
Session evaluation	Depth	5.70 ± 1.0	5.15 ± 0.9	1651.5	0.005 *
	Smoothness	6.45 ± 0.6	5.81 ± 0.9	1803.0	0.000 **
Post-session mood	Positivity	6.26 ± 0.9	5.49 ± 1.1	1783.0	0.000 **
	Arousal	3.03 ± 0.9	3.55 ± 0.9	873.5	0.009 *

Note. Data are presented as mean ± standard deviation. *p* values for differences between groups come from an analysis of Mann-Whitney U test. * Statistically significant at *p* < 0.01. ** Statistically significant at *p* < 0.001.

## Data Availability

The data presented in this study are available on request from the corresponding author.

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
