# Peer review of "Effectiveness of Simulation-Based Empathy Enhancement Program for Caregivers (SEE-C) Evaluated by Older Adults Receiving Care"

_ijerph, 2021, doi:10.3390/ijerph18157802_

Round 1

Reviewer 1 Report

SUMMARY

The researchers used an experimental design to test whether a simulation-based training program enhanced the empathy of caregivers, as measured by older adults receiving the care. Some of the caregivers received the simulation-based empathy enhancement program while others did not. The older adults were randomly assigned to a caregiver who received the simulation-based program or a caregiver who did not. After an interview with the caregiver, the older adults completed a survey to evaluate the interview session and measure their mood positivity and arousal. As expected, older adults who were interviewed by simulation-trained caregivers gave the interview session higher marks and reported higher levels of positive mood than those who were interviewed by caregivers without the training. The reverse was true for arousal. These results should inform the caregivers’ training and help caregivers deal with the burgeoning older adult population.

EVALUATION

I appreciate the care with which the study was designed, and that it measured the effects of training not on the caregivers, but on the recipients of the care. The results are clear except for the fact that there were age differences between the two groups—this is a serious confound. I offer some suggestions for each section of the manuscript to improve the background of the study and its analysis and discussion.

  1. The study needs to be contextualized better in the introduction.

It should address more clearly what the problem is and why enhancing empathy of caregivers would improve the problem. In other words, why is increased empathy needed? What is involved with typical training programs? Who has been trained with these programs? How is empathy measured? I suggest moving part of the discussion to the introduction to address these issues up front (pp. 6-7, lines 209-239).

  1. The study itself needs more explanation.
    1. The last paragraph in the introduction should have more information about the design of and predictions for the study. For example, “If the training has an effect on older adults’ well-being, then ….”
    2. In 2.1 Research design, put the design first, then describe the training. What training did the other caregivers receive?
    3. What was the procedure for the interview session? What was the interview about?
    4. Arousal is defined in the manuscript as the extent of feeling active and excited (p. 4, line151). However, arousal can be both positive and negative. Why did the authors assume that arousal would be positive?

  1. The age difference between the two groups needs to be addressed in the statistical analysis.

I understand why nonparametric tests were used, but the age difference is a serious confound. I suggest trying different transformations on the data (e.g., 1/x) to see which one produces relatively normal distributions. If one can be found, then age can be used as a covariate.

  1. In addition, the data in Figure 2 raised a few questions that should at least be mentioned.
    1. Two of the measures reached ceiling level for the experimental group (smoothness, positivity). I don’t see this is a serious problem because there were still statistically significant differences between the experimental and control groups.
    2. The description of the whisker box (Figure 2 description) needs to include an explanation of the data points beyond the highest and lowest values (i.e., smoothness and arousal for the control group).
    3. Why was there so much variability in the arousal measure for the control group?

  1. The discussion should compare the current findings with findings from previous studies described in the introduction. This would produce a stronger sense of the importance of the current study and where it fits within current knowledge.

Author Response

Thank you for your hard work.
The review report is written in word and uploaded.
Please see the attachment.

Reviewer 2 Report

Dear Authors,

the effectiveness measure of a simulation based empathy enhancement program for direct care workers of older adults living alone by evaluating older adults who received home visiting care by trained direct care workers through the simulation-based is very interesting, both from a scientific and ethical point of view. 

However, I have some suggestions for improving the quality of the paper:

Methods: it is essential to define very clearly which method is used and the reasons for the choice. Different methods lead to different results?

Discussion: as for the method this section is very important. Is it possible to compare the results obtained with other studies?What are the main considerations that can be made?

Author Response

(The authors gave the same response as above.)

Reviewer 3 Report

“The Effectiveness of Simulation-based Empathy Enhancement Program for Caregivers of the Elderly (SEE-C) Measured by the Evaluation of Older Adults Receiving Care” is well organized and easy to read.

Some specific comments/suggestions are presented below:

Title  - The title is too long. A possible suggestion: “Effectiveness of Simulation-based Empathy Enhancement Program for Caregivers (SEE-C) Evaluated by Older Adults Receiving Care”

Lines 23 to 25 – From this work since the evaluation was made by the older adults receiving care I suggest to keep only the conclusion “…raising the satisfaction of the interviewee.” and remove “…in improving the capacity of the interviewer…”.

Lines 100 to 102 – Refer the statistic test used and the p-value.

Section 2.4 – I suggest to add two sub-topics in order to distinguish the measurement of the health status of each participant and the evaluation of the session (and use the same sub-topics in the results section).

Table 1 – p-value indicates difference in age between the two groups and this is not mentioned in the text in the results section (only at the end of the discussion). Besides this fact an average age of the two groups together is presented… Since there is the need to use non-parametric test please explain why not presenting the median values and inter-quartile range instead.

Table 2 – redundant information (already in the text the results of the Mann-Whitney U test and descriptive statistics in Figure 2)

Figure 2 – If whiskers extend to the largest (LS) and smallest (LI) data points there shouldn’t be outliers (represented by the small dots and not referred in the text). Is it LI=Q1-c.IQR and LS=Q3+c.IQR, being Q1 – 1st quartile; Q3 – 3rd quartile, c – a constant usually equal to 1.5, IQR – inter-quartile range)?

Author Response

(The authors gave the same response as above.)

Round 2

Reviewer 2 Report

Dear Authors,

the paper has been improved. The recommendations of the reviewers were followed.

Now the paper can be accepted